# Exploring the Influence of Sociodemographic Characteristics on the Utilization of Maternal Health Services: A Study on Community Health Centers Setting in Province of Jambi, Indonesia

**DOI:** 10.3390/ijerph19148459

**Published:** 2022-07-11

**Authors:** Herwansyah Herwansyah, Katarzyna Czabanowska, Stavroula Kalaitzi, Peter Schröder-Bäck

**Affiliations:** 1Department of International Health, Care and Public Health Research Institute (CAPHRI), Faculty of Health, Medicine and Life Sciences, Maastricht University, P.O. Box 616, 6200 MD Maastricht, The Netherlands; kasia.czabanowska@maastrichtuniversity.nl; 2Public Health Study Program, Faculty of Medicine and Health Sciences, Universitas Jambi, Telanaipura, Kota Jambi 36122, Indonesia; 3Department of Health Policy Management, Institute of Public Health, Faculty of Health Sciences, Jagiellonian University, 31-007 Krakow, Poland; 4Department of Global Health, Richard M. Fairbanks School of Public Health, Indiana University, Bloomington, IN 47405, USA; valiakalaitzi@gmail.com; 5Institute of History and Ethics of Police and Public Administration (IGE), University of Applied Sciences for Police and Public Administration, North Rhine-Westphalia, 52068 Aachen, Germany; peter.schroeder-baeck@hspv.nrw.de

**Keywords:** maternal health services, community health centers, sociodemographic

## Abstract

The Maternal Mortality Ratio in Indonesia has remained high, making it a national priority. The low utilization of maternal health services at community health centers is considered to be one of the reasons for poor maternal health status. This study aims to assess the influence of sociodemographic factors on utilization of maternal health services. The analysis was completed using binary and logistic regression to examine the association between sociodemographic variables and maternal health services utilization. A total of 436 women participated in the survey. In the multivariable analysis, age, education, ethnicity, parity status, distance to health centers and insurance ownership were associated with the utilization of maternal health services. Ethnicity (OR, 2.1; 95% confidence interval, 1.4–3.3) and distance to the CHC (OR, 0.5; 95% confidence interval, 0.3–0.8) were significantly associated with ANC visits. The association between parity and place of delivery was statistically significant (OR, 0.8; 95% confidence interval, 0.5–1.4). A positive association between basic health insurance ownership and PNC services was reported (OR, 0.3; 95% confidence interval, 0.1–0.6). Several sociodemographic factors were positively associated with the utilization of maternal health services at the CHCs. The required measures to improve the utilization of maternal health services at the CHCs level have to take into consideration the sociodemographic factors of reproductive age women.

## 1. Introduction

Maternal death remains a global health problem, particularly in many Low and Middle-Income Countries (LMICs). Most of these countries are continuing to prioritize programs to reduce maternal mortality in order to achieve Sustainable Development Goal (SDG) target 3, which aims to reduce the maternal mortality ratio to less than 70 per 100,000 live births by 2030 [1]. Despite the fact that some countries have achieved substantial improvements in maternal mortality reduction ratio, the outcomes remain poor in most LMICs [2]. Inadequate health expenditure, limited access to the nearest healthcare facilities, and lack of skilled health workers and provision of good-quality care are only some of the key challenges facing these countries to improve maternal health [3,4,5].

Indonesia is one of the biggest LMICs, with a rapidly developing economy, yet the health sector is still a major concern, including the high maternal mortality ratio (MMR). The country has one of the highest maternal mortality ratios among neighboring countries in Southeast Asia [6]. Data show that the mortality ratio is 359 per 100,000 live births [7]. This figure is still far from the government target of 232 per 100,000 in 2024 [6,8,9].

Clearly, it has been challenging for Indonesia to reduce the maternal mortality target for the last 15 years. Several key strategies for reducing MMR have been implemented, including the provision of comprehensive Maternal Health Services (MHS) at Community Health Centers (CHCs) with the aim to provide equal services for communities in both rural and urban areas [10,11]. Maternal health services in the primary healthcare setting consist of Antenatal Care (ANC), health institution delivery, and Postnatal Care (PNC) [2]. Receiving adequate services at CHCs has been the most relevant intervention to minimize maternal death in Indonesia [12]. The CHC is the main entry point into the basic healthcare system and mainly delivers preventive and health promotion services. However, ensuring women use these services is an extremely challenging task, especially for poor areas where health infrastructure and skilled health professionals are often limited [13,14]. Additionally, access to public health facilities poses a significant challenge for improving the utilization of maternal health services, particularly among the underprivileged populations [15]. The low utilization of MHS also occurs in urban areas where women generally visit higher-level health facilities to receive maternal health services [16]. These obstacles have received rather poor attention from policy- and decision-makers when developing and implementing interventions to enhance maternal health service utilization at the CHC level. Evidence suggests that there are several factors associated with the low level of utilization of maternal health services, including socioeconomic and political context, community context, and the health system [17,18,19].

The varying degree of utilization of maternal health services at the primary healthcare level is caused by a range of factors that can be categorized as structural and intermediary determinants [20]. The structural factors are generally related to producing health inequities by generating social stratification in the community. The structural factors include socioeconomic and political contexts, structural mechanisms, and socioeconomic position. The intermediary determinants produce differential health-compromising conditions, which include individual-level, health system, and community contextual factors. The structural and sociocultural factors are equally important to address the causes underlying maternal mortality in the community [21].

The Province of Jambi is one of the small provinces in Indonesia where the MMR remains high (96 per 100,000 live births) compared to other provinces in the country [8,22]. Although the MMR is still under the national figure, this ratio is supposed to be lower. Hence, this is still considered a major concern for the provincial government. Taking into consideration that there is limited evidence in the field of utilization of maternal health services at the CHC level in Indonesia in general and in the Province of Jambi in particular, in response to maternal health outcomes, this study focuses on exploring the utilization of maternal health services in the Province of Jambi. The study aimed to gain deeper insights on potential gaps and room for improvement in the utilization of maternal health services at the CHC level by exploring socioeconomic and demographic factors associated with the utilization of maternal health services at the primary healthcare level in the region of Jambi. The study attempted to contribute to an effective dialogue with stakeholders and policy-makers on improving the utilization of maternal health services and reducing the MMR. The study followed a patient-centered approach, applying a quantitative research method design.

## 2. Materials and Methods

### 2.1. Study Design and Setting

This was a cross-sectional and descriptive study with a quantitative approach that measured the outcomes and exposures in the study population at the same time [23]. The study was conducted in three selected regencies of Jambi Province, namely Municipality of Jambi (MMR 29), Municipality of Sungai Penuh (MMR 68) and Merangin Regency (MMR 70). The regencies were selected purposively based on specific criteria: (1) they represent western, central and eastern regions, (2) they have a significance difference in maternal mortality, (3) the populations in the selected regencies are relatively larger than other regencies across the Province, and (4) all CHCs were easy and convenient to access. The study was conducted from 1 February to 30 April 2021 in 31 CHC working areas in the above-mentioned selected regencies (Figure 1).

### 2.2. Study Population and Data Source

The study population of this research included reproductive age women who had given birth at least once and resided in the area for the last 12 months. Data of reproductive age women were obtained from the maternal cohort book provided by midwives at the CHCs.

### 2.3. Sample Size Determination and Sampling Procedure

The number of respondents required for the study was calculated using Slovin’s formula [24].
n = N/(1 + N e2)
where N is the expected population size (40,813), and e is the error tolerance (0.05). This study aimed to have a minimum target sample size of 396 respondents based on the formula. Additional respondents (10% of the minimum target sample size) were included in this study due to a possibility of selective non-response and drop out, as well as to ensure sufficient respondents at the various CHCs. The final sample size was estimated to be 436 respondents.

Then, 31 CHCs out of 58 were selected considering the limited access to the CHCs due to COVID-19 pandemic restrictions. A simple random sampling technique was employed to recruit and invite the women to participate in the community-based study. An updated list of reproductive age women in the CHC working areas was obtained from the Department of Health in each regency and used as a sampling frame. The sample size was allocated to each selected CHC proportional to the number of women who met the inclusion criteria. Finally, the respondents were randomly selected based on the prepared list until the required sample size was achieved (Figure 2).

### 2.4. Data Collection Procedure

The reproductive age women’s data were collected by the sociodemographic questionnaire in Bahasa Indonesia language. The questionnaire was developed, examined and approved by experts in the field. The questionnaire was piloted on ten women outside the study setting before the commencement of the actual survey. The questionnaire was filled in by the participants; at the same time, they were asked if they understood the questionnaire and about the clarity of items in the questionnaire. After making some revisions to the questionnaire, it was distributed to the respondents by the research assistants. During three months in 2021, research assistants visited the CHCs in three regions. The assistants collected the data by both visiting households and/or waiting for the respondents at the CHCs. The daily activities of the research assistants were supervised by the research team.

### 2.5. Description of Study Variables

To assess the sociodemographic factors of maternal health services utilization, three outcome variables were considered: adequate Antenatal Care (ANC) visits to CHCs during pregnancy, delivery by skilled health personnel and in the primary healthcare facilities, and Postnatal Care (PNC) visits to CHCs within 42 days of delivery. The independent variables included demographic characteristics (women’s age, ethnicity, parity and family type). The socioeconomic characteristics included household income (low income if they earned less than IDR 3 million, and high income if they earned more than or equal to IDR 3 million), religion, educational status, occupation, health insurance and distance to health facilities.

### 2.6. Data Analysis

Data from respondents were checked and entered into Microsoft Excel. Then, the data were exported and analyzed using SPSS (IBM SPSS Version 25, Armonk, NY, USA). The survey dataset was calculated to generate descriptive data, and then was used to calculate the prevalence of maternal health services utilization. The analysis was followed by binary regression to measure the association between the outcome and predictor variables of the study (Hess and Hess, 2019). The variables that met the requirements in the multiple logistic regression model were calculated to analyze the factors influencing ANC visits, delivery and PNC visits. The results of the regression models were presented as odds ratio (OR) and 95% confidence interval (CI). Statistical significance was assumed at *p* < 0.05 [25].

## 3. Results

Table 1 presents the sociodemographic characteristics of the respondents. The majority of respondents were young and middle-aged women, Malay ethnicity, and Muslim. Ninety percent had high education status, with 54% of women having at least a university degree and 36.2% of them graduated from senior high school. In terms of household income, almost two-thirds of the respondents (59.4%) were earning more than IDR 3,000,000 per month. About 57.8% of women lived with their spouses and children. About half of the women were working women (52.7%), and had given birth to at least two children (62.2%).

Regarding maternal health services utilization (Table 2), most (81.0%) of the study participants reported having attended an antenatal care (ANC) visit during their last period of pregnancy. However, only 36% of them had attended the recommended ANC visits during pregnancy. A large proportion (64.7%) of the mothers reported they had delivered at private health facilities, including private midwife clinics and private clinics. More than 50% of women live more than two kilometers away from the nearest community health center. The majority of respondents were covered by the National Health Social Security Agency for health insurance (92.9%).

Table 3 presents the utilization of maternal health services at community health centers by sociodemographic characteristics of the respondents. Overall, 36%, 14% and 35% of the respondents had ≥4 ANC visits, government institutions delivery and PNC within 42 days of delivery, respectively.

Table 4 depicts multivariate logistic regressions for assessing the sociodemographic characteristics associated with the utilization of maternal health services at community health centers. Compared with women aged over 35 years old, women aged between 24 and 35 years old were less likely to have had ≥4 ANC visits and received PNC services at the community health centers. Muslim and Malay women were more than 2 times more likely to receive 4 or more ANC visits compare to non-Muslim and non-Malay women.

Ethnicity, age, education, and parity have a positive influence on the utilization of government health facilities for delivery. Compared with the non-Malay group, Malay women were almost 3 times more likely to have adequate ANC. Women who delivered a child for the first time and younger women were more likely to deliver in the government health facilities than older women. The odds of delivery in the government health facilities were 2.9 times higher among women living with extended families compared with those from nuclear families.

In the case of PNC check-ups, distance to the CHC and health insurance have a significant influence on the PNC visits at the CHCs. The likelihood of having PNC was higher by 1 time among women who lived close to the CHCs compared with those who were living further away from the CHCs. Women with public health insurance were less likely to use PNC, while women who were not covered by the insurance were more likely to have PNC visits within 42 days of delivery.

## 4. Discussion

Although national and regional efforts to improve the utilization of maternal health services at the primary healthcare level have been implemented, a substantial proportion of reproductive age women are not seeking adequate services (delivery at CHCs and PNC) in the investigated three regions in the Province of Jambi. However, significant progress has been made in terms of ANC visits at the community health centers, particularly for the first visit. A high ANC coverage is associated with the government’s new policy, called the triple elimination initiative. The triple elimination initiative of mother-to-child transmission of diseases requires all pregnant mothers to receive ANC services at the CHCs [26,27]. The program is mainly aimed at early detection of disease during pregnancy and increasing the number of first ANC visits at the CHCs. Moreover, pregnant women will receive the maternal health handbook at their first ANC visits to CHCs, which cannot be found at other health facilities. Since the utilization of maternal health services is associated with maternal mortality, the government needs to further improve maternity care in all aspects of services, including ANC, delivery care and PNC.

This study has identified a number of sociodemographic characteristics of reproductive age women for maternal health services utilization at CHCs. Multivariate analysis in this study found that women with low educational attainment were more likely to have ANC and deliver in community health facilities compared with women who were well-educated. This finding is consistent with research findings from a study conducted in another part of Indonesia. The study found that low education women tended to visit community health centers for ANC, whereas the highly educated group chose other health facilities [28]. In other LMICs, for instance Cambodia, urban and well-educated women have greater awareness to access services in non-CHC facilities, whereas rural and uneducated women often visit CHCs since there are no other options to receive maternal health services apart from the CHCs [29]. Women with high education degrees are able to recognize their individual rights to health and make informed decisions when seeking health services [30].

In addition, the findings of this study confirmed that non-working mothers utilize the CHCs’ service for ANC visits and delivery more often than working women. The government of Indonesia has developed a program to provide free maternal health services at the CHC level for all groups of women. However, the services are not entirely utilized. In big cities, women generally hold formal and informal jobs instead of being housewives. Limited time to visit is one of the possible reasons for working women not using the maternal health services at the CHCs. Women tend to receive the service from private clinics or hospitals, where scheduling the visit appointments is flexible. Moreover, working mothers can afford to pay for ANC and delivery services at private health institutions by out-of-pocket payments. Similar studies in Timor-Leste and Nepal identified that financial hardship has been an issue for unemployed women to obtain adequate maternal health services [31,32]. The governments addressed the problem by implementing programs to provide free access to maternal health services at the primary healthcare level.

This study revealed that all of the respondents reported having delivered their last child in health facilities. The majority of women opted to have a normal delivery at private clinics rather than delivery at CHCs. The inpatient community health centers generally provide free basic maternal health services for those who live in the CHC’s area. The CHCs have 24/7 emergency services with 24 h emergency midwives available and a relatively greater number of better-equipped facilities. However, the utilization of these services is still considerably low. A possible reason behind this fact could be that more mothers might have health insurance to cover normal delivery at private clinics and hospitals. This is also associated with the excellent services offered by the private institutions, which is considered more convenient than the services at the CHCs. Similar findings from two studies conducted in Myanmar [33] and the Philippines [34] showed that pregnant mothers preferred to give birth either at hospitals or private clinics. It was reported in the studies that the likelihood of delivering in the hospitals or clinics was greater when the women were covered by health insurance. Additionally, the women felt comfortable receiving comprehensive treatments and services compared to the community health centers.

With regard to the age of the study participants, this study shows that the majority of young women went to the government health institutions for delivery. A similar finding was observed in a study in Vietnam, where the average age of women using the CHCs was 26 years old [35]. Other studies in Southeast Asian countries, such as Indonesia [36] and Brunei Darussalam [37], reported similar findings that the majority of women opted for public health centers for the place of delivery. The possible explanation for the high proportion of young women might be due to receiving positive impressions during the first ANC visit, and developing trust for the health providers at the CHCs. As a result, the midwives are able to influence the young pregnant women on the decision on place of delivery [38].

This study found that PNC visits to the CHCs were significantly low. Most of the women who experienced caesarean delivery received PNC services from the CHCs’ midwives within 42 days after delivery. The CHCs’ health providers have made great efforts to raise women’s awareness about the importance of PNC for both normal and caesarean delivery. Although the health workers had carried out health promotion on postnatal services during the pregnancy period, the women’s interest in receiving PNC services is still low. Similar findings from other developing countries found the utilization of postnatal care services was low, accounting for less than a half of total respondents [17,39,40].

PNC services are crucial for the health of both mothers and newborns. Studies have shown that the utilization of PNC services has numerous impacts on major causes of maternal mortality among the population [41,42]. A study in Indonesia identified potential barriers to the utilization of postnatal care services. Factors included family influences, low health literacy on PNC, and sociocultural beliefs and practices [43]. Improving maternal health through adequate PNC services is one of the key strategies to reduce maternal mortality. Hence, the low utilization of PNC services has to be addressed through various actions.

The results from both bivariate and multivariate analyses confirmed several characteristics of the women in terms of utilization of maternal health services at the CHCs. Ethnicity and distance to the CHC were significantly associated with ANC visits. The study shows that the respondents belong to the Malay ethnicity and live close to the CHCs—70% and 45%, respectively. This finding is consistent with another study that ethnicity influenced the use of maternal health services [44]. Nowadays, Malay women are open to accepting the changes in the health system. A recent study on community perception of healthcare services shows that women from various ethnicities are aware of the importance of seeking healthcare in health facilities [45]. Pregnant women are able to receive maternal health services, since the health facilities are widely available and accessible. Particularly in urban areas, there are a range of options to receive maternal health services, including public and private institutions. In Indonesia, a community health center generally serves a population in one district, and is supported by a small clinic providing basic maternal health services in the community.

The current study also found that the association between parity and place of delivery was statistically significant. This finding was supported by other studies in India, where parity was found to be a strong predictor for the utilization of delivery services at the health facilities [46,47]. A similar study in Kenya indicated that an increase in the number of births tends to reduce the likelihood of delivering in public health facilities [48]. As parity increases, women have experience and knowledge of delivery at the CHCs, and this may be the underlying reason for choosing other health facilities. The women (especially first pregnancy), on the other hand, may have a cultural belief that using the service at the community health center is more convenient and safe than other health facilities.

A positive association between basic health insurance ownership and PNC services was reported in this study. Insurance coverage was generally the underlying reason for mothers using the PNC services at all types of health facilities. Hence, a few mothers prefer to receive the services to private clinics or hospitals other than the CHCs. This contrasts another study [49], which reported that the impact of health insurance on promoting the utilization of PNC services was likely dependent on other characteristics of the women, such as income and education level.

The utilization of services at the primary healthcare level has been considered the most relevant intervention to reduce negative maternal health outcomes as the facility is close to the women. Most importantly, the health providers at the CHCs easily understand the needs and conditions of the community. The government should pay attention to improving the quality of CHCs to attract women’s attention in utilizing maternal health services at the community health centers. Further improvement is required to develop a holistic maternal health program at the CHCs by considering the characteristics of the reproductive age women, in order to meet the users’ expectations.

## 5. Conclusions

This study explored the sociodemographic characteristics of ANC, government institutional delivery and PNC services in three regencies of the Province of Jambi, Indonesia. Several sociodemographic factors, including women age, ethnicity, education level, parity, distance to health facilities and health insurance ownership, were positively associated with the utilization of maternal health services at the CHCs. Despite the remarkable progress in the utilization of the services in general health facilities, community health center-based delivery and postnatal care must be a focus of attention. It was observed that women were less likely to take advantage of the two services, hence interventions should focus on how to improve women’s interest in using the CHCs for delivery and PNC services. Particularly for well-educated and working mothers, a massive campaign for delivery and PNC at CHCs is needed through various platforms, such as social media.

## Figures and Tables

**Figure 1 ijerph-19-08459-f001:**
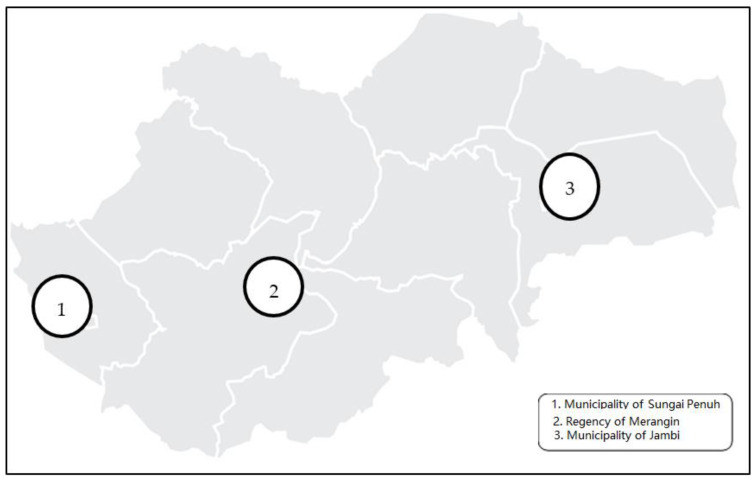
The map of Province of Jambi, Indonesia [22].

**Figure 2 ijerph-19-08459-f002:**
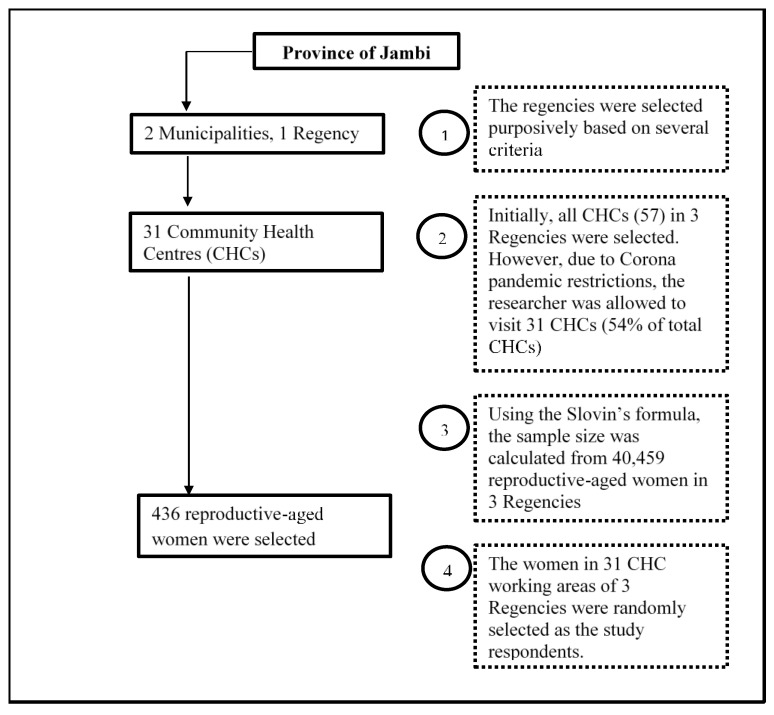
Flow chart of sampling procedures based on this study.

**Table 1 ijerph-19-08459-t001:** Sociodemographic characteristics of women aged 15–49 years, Municipality of Jambi, Municipality of Sungai Penuh, and Merangin Regency, Province of Jambi, 2021 (n = 436).

Characteristics	Categories	Number (%)
**Age during the survey**	<24	75 (17.2)
	24–35	315 (72.2)
	>35	46 (10.6)
**Religion**	Muslim	410 (94.0)
	Non-Muslim	26 (6.0)
**Ethnicity**	Malay	289 (66.3)
	Non-Malay	147 (33.7)
**Educational status**	Elementary school	8 (1.8)
	Junior high school	34 (7.8)
	Senior high school	158 (36.2)
	University	236 (54.1)
**Occupation**	Farming	11 (2.5)
	Government/private employee	131 (30.0)
	Self–employed/entrepreneur	88 (20.2)
	Unemployed/housewife	206 (47.2)
**Type of family**	Extended	184 (42.2)
	Nuclear	252 (57.8)
**Parity**	Primipara	165 (37.8)
	Multipara	271 (62.2)
**Household income**	Low income	177 (40.6)
	High income	259 (59.4)
**Total N**		436 (100.0)

**Table 2 ijerph-19-08459-t002:** Maternal health services utilization of the respondents, Municipality of Jambi, Municipality of Sungai Penuh, Merangin Regency, Province of Jambi, 2021 (n = 436).

Characteristics	Categories	Number (%)
**ANC visit to CHCs**	1 time	83 (19.0)
	2 times	143 (32.8)
	3 times	53 (12.2)
	≥4 times	157 (36.0)
**Place of delivery**	Midwife/Clinic (Private)	282 (64.7)
	Midwife/CHC (Government)	59 (13.5)
	Hospital	95 (21.8)
**PNC visit to CHCs**	Yes	152 (34.9)
	No	284 (65.1)
**Distance to CHCs**	≤2 kilometers	190 (43.6)
	>2 kilometers	246 (56.4)
**Health insurance**	BPJS	405 (92.9)
	Non-BPJS	31 (7.1)
**Total N**		436 (100.0)

**Table 3 ijerph-19-08459-t003:** Utilization of maternal health services at community health centers by sociodemographic characteristics of the respondents, Municipality of Jambi, Municipality of Sungai Penuh, and Merangin Regency, Province of Jambi, 2021 (n = 436).

Variables	Categories	N ANC Visits (%)		N Place of Delivery (%)		N PNC Visits (%)	
		≤3 Times	≥4 Times	Others	Gov/CHC	No	Yes
**Age during the survey**	<24	44 (15.8)	31 (19.7)	49 (13.0)	26 (44.1)	56 (19.7)	19 (12.5)
	24–35	214 (76.7)	101 (64.3)	285 (75.6)	30 (50.8)	211 (74.3)	104 (68.4)
	>35	21 (7.5)	25 (15.9)	43 (11.4)	3 (5.1)	17 (6.0)	29 (19.1)
**Religion**	Non-Muslim	21 (7.5)	4 (3.2)	26 (6.9)	0 (0)	16 (5.6)	10 (6.6)
	Muslim	258 (92.5)	152 (96.8)	351 (93.1)	59 (100)	268 (94.4)	142 (93.4)
**Ethnicity**	Non-Malay	110 (39.4)	37 (23.6)	137 (36.3)	10 (16.9)	91 (32.0)	56 (36.8)
	Malay	169 (60.6)	120 (76.4)	240 (63.7)	49 (83.1)	193 (68.0)	96 (63.2)
**Educational status**	Low education	23 (8.2)	19 (12.1)	25 (6.6)	17 (28.8)	32 (11.3)	10 (6.6)
	High education	256 (91.8)	138 (87.9)	352 (93.4)	42 (71.2)	252 (88.7)	142 (93.4)
**Occupation**	Non-worker	127 (45.5)	79 (50.3)	165 (43.8)	41 (69.5)	143 (50.4)	63 (41.4)
	Worker	152 (54.5)	78 (49.7)	212 (56.2)	18 (30.5)	141 (49.6)	89 (58.6)
**Type of family**	Nuclear	161 (57.7)	91 (58.0)	231 (61.3)	21 (35.6)	154 (54.2)	98 (64.5)
	Extended	118 (42.3)	66 (42.0)	146 (38.7)	38 (64.4)	130 (45.8)	54 935.5)
**Parity**	Primipara	108 (38.7)	57 (36.3)	140 (37.1)	25 (42.4)	117 (41.2)	48 (31.6)
	Multipara	171 (61.3)	100 (63.7)	237 (62.9)	34 (57.6)	167 (58.8)	104 (68.4)
**Household income**	Low	104 (37.3)	73 (46.5)	136 (36.1)	41 (69.5)	124 (43.7)	53 (34.9)
	High	175 (62.7)	84 (53.3)	241 (63.9)	18 (30.5)	160 (56.3)	99 (65.1)
**Distance to CHCs**	≤2 KM	105 (37.6)	85 (54.1)	170 (45.1)	20 (33.9)	107 (37.7)	83 (54.6)
	>2 KM	174 (62.4)	72 (45.9)	207 (54.9)	39 (66.1)	177 (62.3)	69 (45.4)
**Health insurance**	Non-BPJS	20 (7.2)	11 (7.0)	31 (8.2)	0 (0)	11 (3,9)	20 (13.2)
	BPJS	259 (92.8)	146 (93.0)	346 (91.8)	59 (100)	273 (96.1)	132 (86.8)
**Overall**		64.0	36.0	86.5	13.5	65.1	34.9

**Table 4 ijerph-19-08459-t004:** Multivariate logistic regression for sociodemographic factors associated with the utilization of maternal health service at community health centers in Province of Jambi, 2021.

Variables	≥4 Times ANC Visits to CHC		Delivery at the Government Institutions		PNC Visits to CHC within 42 Days	
	OR	95% CI	OR	95% CI	OR	95% CI
**Age**						
**<24**	0.592	(0.282–1.241)	7.605	(2.150–26.901)	0.199	(0.090–0.440)
**24–35**	0.396	(0.212–0.742)	1.509 *	(0.441–5.159)	0.289 *	(0.152–0.550)
**>35 (Ref)**	1.00		1.00		1.00	
**Religion**						
**Non-Muslim (Ref)**	1.00		1.00		1.00	
**Muslim**	2.474	(0.914–6.697)	0.000	(0.000–0.000)	0.848	(0.375–1.917)
**Ethnicity**						
**Non-Malay (Ref)**	1.00		1.00		1.00	
**Malay**	2.111 *	(1.360–3.277)	0.358 *	(0.175–0.728)	0.808	(0.535–1.222)
**Education**						
**Low education (Ref)**	1.00		1.00		1.00	
**High education**	0.653	(0.343–1.240)	0.175 *	(0.088–0.351)	1.803	(0.861–3.776)
**Occupation**						
**Non-worker (Ref)**	1.00		1.00		1.00	
**Worker**	0.825	(0.558–1.220)	0.342	(0.189–0.617)	1.433	(0.963–2.133)
**Type of Family**						
**Nuclear (Ref)**	1.00		1.00		1.00	
**Extended**	0.990	(0.666–1.470)	2.863	(1.616–5.072)	0.653	(0.435–0.980)
**Parity**						
**Primipara (Ref)**	1.00		1.00		1.00	
**Multipara**	1.108	(0.739–1.661)	0.803 *	(0.460–1.402)	1.518	(1.002–2.300)
**Household income**						
**Low (Ref)**	1.00		1.00		1.00	
**High**	0.684	(0.460–1.017)	0.248	(0.137–0.448)	1.448	(0.963–2.176)
**Distance to CHC**						
**≤2 KM (Ref)**	1.00		1.00		1.00	
**>2 KM**	0.511 *	(0.344–0.760)	1.601	(0.900–2.849)	0.503 *	(0.337–0.749)
**Health Insurance**						
**Non-BPJS (Ref)**	1.00		1.00		1.00	
**BPJS**	1.025	(0.478–2.199)	0.000	(0.000–0.000)	0.266 *	(0.124–0.571)
**Pseudo R^2^**	0.91		0.26		0.11	

Notes: * *p* < 0.05; OR: unadjusted odds ratio, CI: confidence interval; Ref: reference category of the variable.

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
