# Peer review of "Exploring the Influence of Sociodemographic Characteristics on the Utilization of Maternal Health Services: A Study on Community Health Centers Setting in Province of Jambi, Indonesia"

_ijerph, 2022, doi:10.3390/ijerph19148459_

Round 1
Reviewer 1 Report
The article is very well written, the background is logically connected to the methodology of the study. The sampling, data collection, data analyses are all very clear, however the main question still remains, what is new? How does this study contribute to the field?
As authors claim:
‘Evidence suggested that there are several factors associated with the low level of maternal health services utilization, including socio-economic 69 and political context, community context, and health system”
I addition, evidence on barriers to health care utilization among women and characteristics of these women residing in LIC are also well described and analyzed in the literature and that fact is well explained by authors themselves in the article, e.g.:
“The varying degree of utilization of maternal health services at the primary 71 healthcare level is caused by a range of factors that can be categorized as structural and 72 intermediary determinants [20]. The structural factors are generally related to producing 73 health inequities by generating social stratification in the community. The structural fac-74 tors include socio-economic and political contexts, structural mechanisms, and socio-eco-75 nomic position. The intermediary determinants produce differential health-compromising conditions, which include individual-level, health system, and community contextual 77 factors. The structural and socio-cultural factors are equally important to address the 78 causes underlying maternal mortality in the community [21.
The gap in our knowledge regarding the given problem is explained briefly but it is not convincing:
“Hence, this is still being a major 82 concern for the provincial Government. Taking into consideration that there is scarce evidence in the field of the utilization of maternal health services at the CHC level in Indonesia generally and Province of Jambi particularly, this study focuses on exploring the 85 utilization of maternal health services in the Province of Jambi. The study aims to gain 86 deeper insights on the potential gaps and room for improvement in the utilization of ma-87 ternal health services at the CHC level by exploring socio-economic and demographic fac-88 tors associated with the utilization of maternal health services at the primary healthcare 89 level in the region of Jambi, The study will attempt to contribute to an effective dialogue 90 with stakeholders and policy-makers on improving the utilization of maternal health ser-91 vices and reducing the MMR. The study follows a patient-centered approach applying a quantitative research method design.
The discussion part supports the idea expressed above. The findings are similar to previous studies which takes us back to the same question, what is the new knowledge? How does this study enhance our understanding of the problem? Authors could have collected data that could have helped them to produce new knowledge by including variables that were not included in previous studies or collect qualitative data through interviews so we could hear the voices of women.
Author Response
Thank you for giving us the opportunity to submit a revised draft of our manuscript titled Exploring the Influence of Sociodemographic Characteristics on the Utilization of Maternal Health services: A Study on Community Health Centers Setting in Province of Jambi, Indonesia to the International Journal of Environmental Research and Public Health. We appreciate the time and effort that you and the reviewers have dedicated to providing your valuable feedback on my manuscript. We are grateful to the reviewers for their insightful comments on my paper. We have been able to incorporate changes to reflect most of the suggestions provided by the reviewers. We have highlighted the changes within the manuscript. Here is a point-by-point response to the reviewers’ comments and concerns.

Author Response

(The authors gave the same response as above.)

Round 2
Reviewer 1 Report
thanks for your request
most of the responses addressed the questions/ concerns I raised, however, authors could have done the better job with my comment 5
Comment 5:
The discussion part supports the idea expressed above. The findings are similar to previous studies, which takes us back to the same question, what is the new knowledge? How does this study enhance our understanding of the problem? Authors could have collected data that could have helped them to produce new knowledge by including variables that were not included in previous studies or collect qualitative data through interviews so we could hear the voices of women.
authors did compare the study findings to other settings however, in their response they refer to new perspectives and aspects which still have not been explained well. they highlight the similar findings yet new perspectives that have not been discovered in previous literature are still lacking.
Author Response
Dear Respected Editor and Reviewers,  
Thank you for giving us the opportunity to submit a revised draft of our manuscript titled Exploring the Influence of Sociodemographic Characteristics on the Utilization of Maternal Health services: A Study on Community Health Centers Setting in Province of Jambi, Indonesia to the International Journal of Environmental Research and Public Health. We appreciate the time and effort that you and the reviewers have dedicated to providing your valuable feedback on my manuscript. We are grateful to the reviewers for their insightful comments on my paper. We have been able to incorporate changes to reflect most of the suggestions provided by the reviewers. We have highlighted the changes within the manuscript. Here is a point-by-point response to the reviewers’ comments and concerns.
